# Rapid expansion and international spread of M1$_{UK}$ in the post-pandemic UK upsurge of *Streptococcus pyogenes*

Ana Vieira[1,2,3], Yu Wan [1,3,4], Yan Ryan[5], Ho Kwong Li [1,2], Rebecca L. Guy[4], Maria Papangeli[1,2], Kristin K. Huse[1,2], Lucy C. Reeves[1,2], Valerie W. C. Soo [1,2], Roger Daniel[5], Alessandra Harley[5], Karen Broughton[5], Chenchal Dhami[5], Mark Ganner[5], Marjorie A. Ganner[5], Zaynab Mumin[5], Maryam Razaei[5], Emma Rundberg[5], Rufat Mammadov[5], Ewurabena A. Mills[1,2], Vincenzo Sgro[1], Kai Yi Mok[1], Xavier Didelot [6], Nicholas J. Croucher [7,8], Elita Jauneikaite [3,7,8], Theresa Lamagni[3,4], Colin S. Brown [3,4], Juliana Coelho [3,4,5] ✉ & Shiranee Sriskandan [1,2,3] ✉

The UK observed a marked increase in scarlet fever and invasive group A streptococcal infection in 2022 with severe outcomes in children and similar trends worldwide. Here we report lineage M1$_{UK}$ to be the dominant source of invasive infections in this upsurge. Compared with ancestral M1$_{global}$ strains, invasive M1$_{UK}$ strains exhibit reduced genomic diversity and fewer mutations in two-component regulator genes *covRS*. The emergence of M1$_{UK}$ is dated to 2008. Following a bottleneck coinciding with the COVID-19 pandemic, three emergent M1$_{UK}$ clades underwent rapid nationwide expansion, despite lack of detection in previous years. All M1$_{UK}$ isolates thus-far sequenced globally have a phylogenetic origin in the UK, with dispersal of the new clades in Europe. While waning immunity may promote streptococcal epidemics, the genetic features of M1$_{UK}$ point to a fitness advantage in pathogenicity, and a striking ability to persist through population bottlenecks.

Group A Streptococcus (GAS, *Streptococcus pyogenes*) is a human-restricted pathogen causing diseases ranging from sore throat and scarlet fever to more serious invasive infections, including soft tissue infections, pneumonia, and toxic shock, as well as auto-immune sequelae[1]. Although advanced age and specific presentations such as necrotising fasciitis increase the risk of death from invasive infection, the genetic background of *S. pyogenes* strains also contributes to the risk of mortality[2,3] underlining the role of strain genotype and virulence in disease outcome. Among more than 250 recognised *emm* types, the *emm*1 genotype is most frequently associated with invasive infections in high-income countries[4]. *emm*1 strains are considered highly virulent[5,6] and often acquire inactivating mutations in the *covRS* two-component regulator, which de-represses key virulence factors during invasive infection[7]. In the 1980s, *emm*1 emerged as a leading cause of invasive infection following several genomic changes that altered phage content and streptolysin O (SLO) expression, leading to a new clone that spread globally[8].

[1]Department of Infectious Disease, Imperial College London, London, UK. [2]Centre for Bacterial Resistance Biology, Imperial College London, London, UK. [3]NIHR Health Protection Research Unit in Healthcare-associated Infections and AMR, Imperial College London, London, UK. [4]Healthcare-Associated Infections, Fungal, AMR, AMU, and Sepsis Division, UK Health Security Agency, London, UK. [5]Reference Services Division, UK Health Security Agency, London, UK. [6]School of Life Sciences and Department of Statistics, University of Warwick, Coventry, UK. [7]School of Public Health, Imperial College London, London, UK. [8]MRC Centre for Global Infectious Disease Analysis, Imperial College London, London, UK. ✉e-mail: Juliana.coelho@ukhsa.gov.uk; s.sriskandan@imperial.ac.uk

In England, prompt notification and antibiotics are advocated for scarlet fever and invasive GAS (iGAS) infections[9], however guidelines that recommend a non-treatment or delayed treatment approach to sore throat were introduced in 2008, to limit unnecessary use of antibiotics[10]. Unexpectedly large seasonal upsurges in scarlet fever were documented annually in England between 2014-2018[11,12] coinciding with the expansion and recognition of a new lineage of *emm*1 termed M1$_{UK}$ among *S. pyogenes* isolates[5]. M1$_{UK}$ differed from other globally circulating *emm1* strains[8] (hereafter referred to as M1$_{global}$) by 27 signature SNPs and was characterised by increased expression of the scarlet fever toxin, streptococcal pyrogenic exotoxin A (*speA*)[5,6,13]. Two intermediate lineages, M1$_{13SNPs}$ and M1$_{23SNPs}$, that share subsets of the 27 SNPs, were also identified[5,6]. M1$_{23SNPs}$ expresses SpeA at the same level typical of M1$_{UK}$, whereas M1$_{13SNPs}$ does not[6]. By 2016, the M1$_{UK}$ lineage represented 84% of all *emm1* invasive strains in England[5], increasing to 91.5% by 2020[14].

The onset of the COVID-19 pandemic, and implementation of non-pharmaceutical interventions (NPI) to limit SARS-CoV2 transmission triggered a reduction in scarlet fever and iGAS notifications in 2020[12]. However, in late 2022, a highly pronounced out-of-season upsurge in both scarlet fever and iGAS cases was reported in England, with unexpected increase in paediatric pleural empyema and several fatalities[15]. Similar increases in severe paediatric iGAS infections were reported worldwide[16].

In this article, we show that the *S. pyogenes* upsurge in England and Wales was predominantly associated with M1$_{UK}$, a lineage we estimate to have emerged around 2008, and, in particular, three emergent clades that are now widely dispersed. The expansion of M1$_{UK}$ occurred following a bottleneck in growth, likely related to reduced transmission during the COVID-19 pandemic.

## Results
### Trends in *S. pyogenes*-positive samples, England 2016–2023
*S. pyogenes* identified from non-sterile and sterile site samples are recorded through a national laboratory reporting system (Second Generation Surveillance System, SGSS). The typical pattern of seasonal spring-time peaks (Q1-Q2) in *S. pyogenes* infections was interrupted abruptly in April 2020, coinciding with NPI introduced at the onset of the COVID-19 pandemic (Fig. 1). A profound reduction in *S. pyogenes*-positive samples, from both sterile and non-sterile sites, lasted almost

two years, ending in Q1 2022. Following cessation of widespread NPI in February 2022, a delayed seasonal increase in microbiologically-confirmed *S. pyogenes* infections returned in April 2022, subsiding only in Q3 2022, in keeping with the UK summer vacation period. Unexpectedly, a second, exponential increase in *S. pyogenes* samples occurred in Q4 of 2022 (Fig. 1). This marked increase in microbiologically-confirmed infections peaked in week 49, when 8906 non-sterile site and 241 sterile site *S. pyogenes*-positive samples were recorded (Fig. 1), coinciding with increased disease notifications[15,17].

*S. pyogenes* isolates cultured from iGAS cases are submitted to the national reference laboratory for *emm* typing. Between Q1 of January 2017 and Q1 of 2020, *emm*1 was the leading cause of iGAS, responsible for 16-28% of all iGAS cases; *emm*1 dominance was greater in children than adults (Fig. 2). During the period of COVID-19-related NPI, annual iGAS isolates reduced ~6.5-fold in children (274 isolates/year 2017-2019; 44 in 2021) and ~2.5-fold in adults (1944 isolates/year 2017-2019; 785 in 2021) (Fig. 2). The proportion of iGAS isolates that were *emm*1 also reduced significantly (p < 0.001), to less than 8% of all iGAS cases. From Q1 of 2022, *emm*1 then showed a sustained quarterly increase in frequency, peaking in Q1 of 2023. For over nine months, *emm*1 accounted for > 50% of all iGAS cases, coinciding with the period of upsurge (Fig. 2). Indeed, *emm*1 was the only genotype to expand significantly during this time, increasing from 20% to 55%. In children (< 15 years), this increase was more apparent; *emm*1 accounted for 60% and 70% of iGAS in the same period (Fig. 2).

### Population genomics of *emm*1 *S. pyogenes* strains comprising the upsurge
To investigate any genetic basis for the increase in *emm*1 iGAS cases, genomes of all 1092 iGAS *emm*1 isolates submitted to the reference laboratory from January 2022 to March 2023 were whole genome sequenced. Phylogenetic analysis revealed clustering of *emm*1 genomes into expected lineages. The vast majority (1001/1092, 91.8%) of isolates were M1$_{UK}$, 4.1% (44/1092) were derivatives of M1$_{UK}$ having lost the phi5005.3 phage (and therefore lacking the phage portal protein SNP that is typical but not essential to M1$_{UK}$) and 4.2% (46/1092) were M1$_{global}$. Taken together, 95.7% of all *emm*1 strains from the 2022/2023 upsurge period were M1$_{UK}$ or a 26SNP derivative thereof, representing overall expansion of the lineage since 2020[14] (Fig. 3A). Isolates from 2022/2023 were further compared to 723 *emm*1 iGAS strains

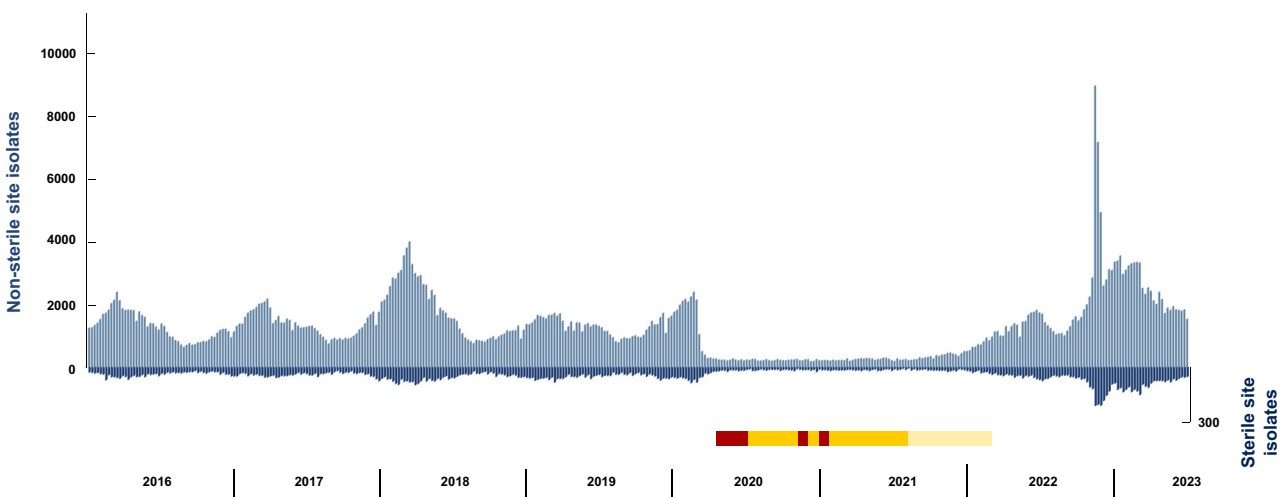

**Fig. 1 | Trend in *S. pyogenes*-positive samples, England 2016–2023.** Data show absolute numbers of weekly *S. pyogenes*-positive samples from non-sterile sites (light blue bars, left hand, positive axis) and sterile sites (dark blue bars, right hand, negative axis) recorded by the Second Generation Surveillance System (SGSS) in England, by week and by year. Timing of non-pharmaceutical interventions (NPI) related to COVID-19 in England is indicated by the horizontal bar: red, lockdown periods; orange, legally enforced NPI including no mixing; yellow, non-severe NPI. Schools were closed during lockdown periods and between the two later lockdown periods except for children of key workers and vulnerable children. Source data are provided as a Source Data file Fig. 1.

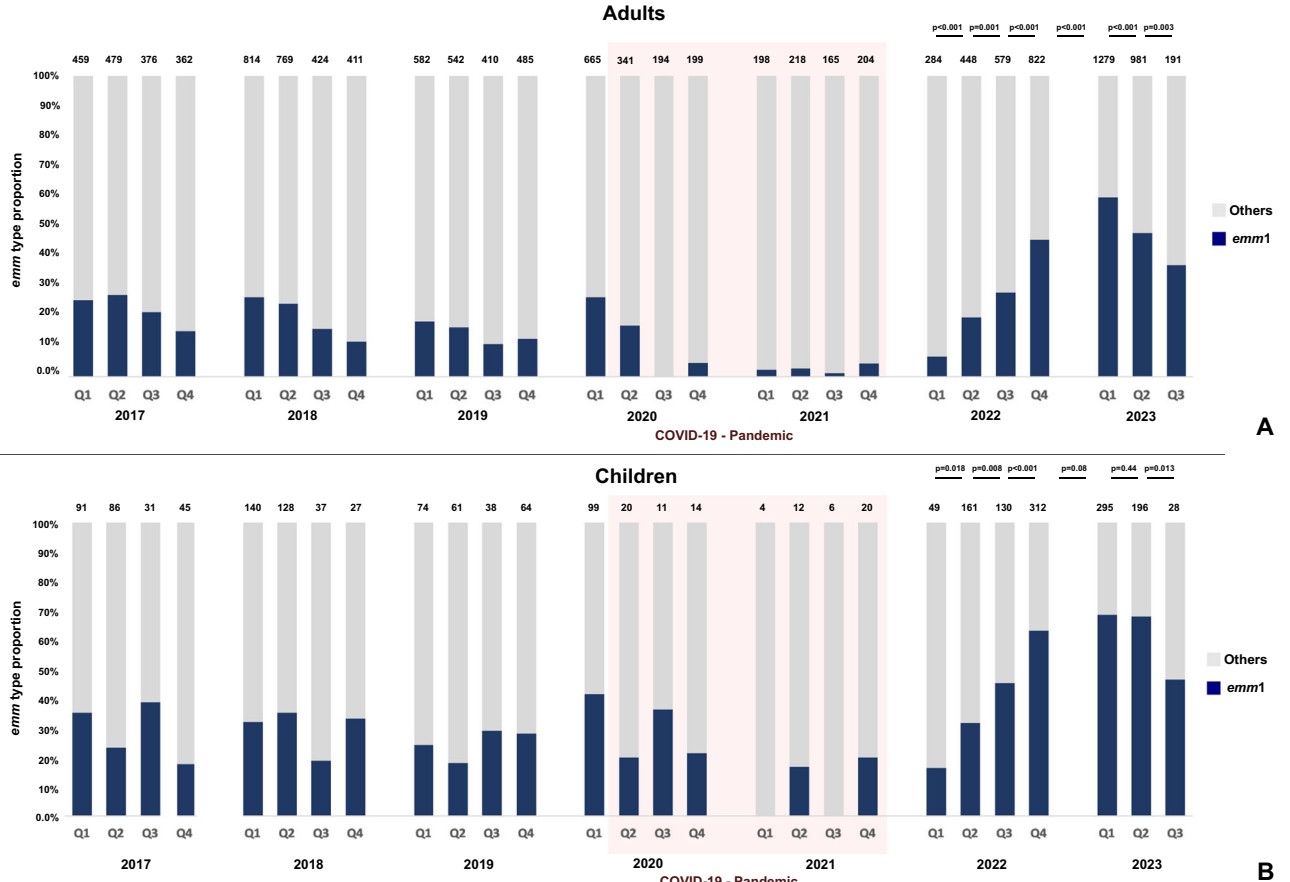

**Fig. 2 | Contribution of *emm*1 *S. pyogenes* to invasive group A streptococcal (iGAS) infections 2017–2023.** *emm*1 isolates are shown as proportions of the total number of isolates from iGAS cases submitted to and genotyped at the national reference laboratory for each quarter of each year. **A** adults (≥15 years); **B** children <15 years. The total number of isolates from iGAS cases received by the reference laboratory and genotyped in each quarter are shown on top of each bar; *emm*1 proportions are shown in navy blue. Pink shaded region highlights the period of COVID-19 non pharmaceutical interventions. Q1, January-March; Q2, April-June; Q3, July- September; Q4, October-December. Statistical analysis applied to 2022-2023: one-tailed proportion test of *emm*1 from Q1 2022 to Q3 2023 (p-values indicated in the figure). Source data are provided as a Source Data file Fig. 2.

sequenced in the same reference laboratory between 2013-2021 to determine evidence for recent genomic change. Phylogenetic analysis of these 1815 *emm*1 *S. pyogenes* genomes associated with iGAS showed M1$_{UK}$ isolates from 2022/2023 to be broadly distributed across the pre-existing M1$_{UK}$ population, with three emergent dominant clades and several small clades formed almost exclusively of isolates from 2022/2023 (Fig. 3A). Three clades accounted for over half (54.8%) of all M1$_{UK}$ from 2022/2023. Clade 1 comprised 123 invasive strains exclusively from 2022/2023 and was characterised by two SNPs (Supplementary Table 1). Clade 2 comprised 166 invasive strains exclusively from 2022/2023 and was characterised by 6 SNPs, including three non-synonymous mutations (in *sic*1.01, *pyrC* and M5005_Spy1146). Clade 3 comprised 284 strains from 2022/2023, plus a single strain collected in February 2020, and was defined by 3 non-synonymous mutations (in *xerD*, *huTu* and *secA*). Clade 3 was enriched by invasive strains collected in southern England (70%), consistent with regional transmission. In contrast, Clades 1 and 2 had similar proportions of strains from northern (26% and 35%), southern (43% and 35%), and central regions including Wales (23% and 28%) consistent with a wider national outbreak (Fig. 3A). The average genetic distance between any two strains from Clade 1 was just 2 SNPs, while for Clades 2 and 3, the average was just 3 SNPs (Supplementary Table 2). The low diversity was consistent with rapid emergence and dispersion through the year and across the country from a recent common ancestor.

Among the 1815 *emm*1 genomes associated with iGAS from 2013-2023, the clinical sources of isolates were known for most strains:

67.7% (1229/1815) were blood isolates; 6.9% (125/1815) were lower respiratory tract isolates, of which 71.2% (89/125) were pleural sample isolates, indicative of empyema (Supplementary Table 3). Overall, a higher proportion of M1$_{UK}$ (5.0%) isolates were associated with pleural samples compared to M1$_{global}$ (2.6%), in particular Clade 3 (8.4%) (Supplementary Table 4). Considering only diseases occurring in 2022/2023, inter-lineage differences were not significant, however M1$_{global}$ isolate numbers were very low (Supplementary Table 4). Pleural sample isolates were notably more frequent at the time of the upsurge. Despite the notable impact of the upsurge on children, no single clade was uniquely associated with a specific age group, and closely related strains (<3 SNPs apart) caused invasive infections in both adults and children (Supplementary Fig. 1).

The average pairwise distance within M1$_{UK}$ increased from 16 SNPs in 2013-2021 to 22 SNPs in 2022/2023, while the average pairwise distance within the M1$_{global}$ lineage increased from 39 SNPs in 2013-2021 to 55 SNPs in 2022/2023 (Supplementary Table 2). Despite the recent increase in the genetic diversity of both lineages (M1$_{global}$ and M1$_{UK}$), M1$_{UK}$ showed greater genomic stability (point mutations) than M1$_{global}$. Most mutations (excluding the 27 M1$_{UK}$ signature SNPs) were unique to individual strains outside the main clades (Supplementary Fig. 2) consistent with a rapid population size expansion. The four indels previously reported[13] were present in 99% of M1$_{UK}$ isolates but were not lineage specific (Supplementary Data 1).

Recombination and pangenome analyses showed little evidence of gain or loss of transferable elements between M1$_{UK}$ and M1$_{global}$, and

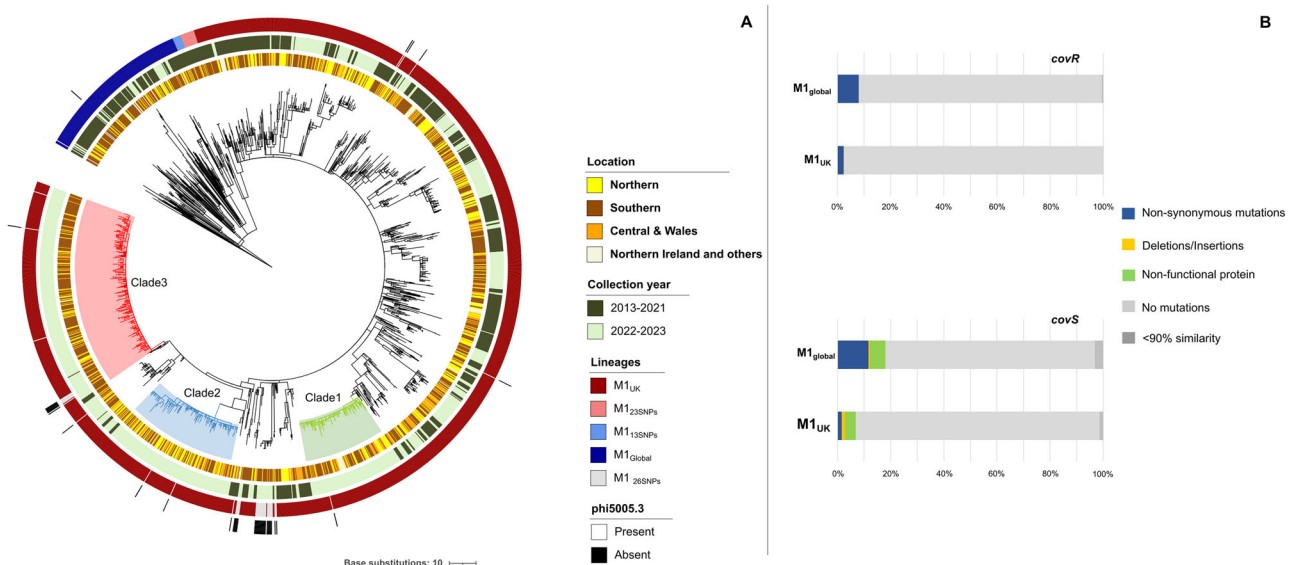

**Fig. 3 | Genetic analysis of 1815 *emm1 S. pyogenes* isolates from invasive group A streptococcal (iGAS) infections 2013-2023. A** Phylogenetic tree comprising sequenced *emm1* isolates associated with invasive infections (iGAS) from 2013-2023 sequenced at reference laboratory: Maximum likelihood phylogenetic tree constructed from 278 core SNPs (excluding recombination regions) extracted after mapping 1815 *emm1* isolates to the MGAS5005 reference genome. The tree was drawn in a circular layout and rooted on outgroup genome NCTC8198. Bars in concentric circles represent (from inside to outside) regional location of isolate; collection period (pre-upsurge 2013-2021 or upsurge 2022-2023); *emm*1 lineage, and presence/absence of the phi5005.3 phage. Regional data have been grouped for purpose of data visualisation as follows: Northern (North-East England, North-West England, Yorks & Humber); Central and Wales (East Midlands, West Midlands, Wales); Southern (South-East England, South-West England, London); and Northern Ireland and others (comprises regions with less than 5 isolates including

Scotland, Eire, Jersey, Malta). **B** Frequency of *covR* and *covS* non-synonymous and other mutations within M1$_{UK}$ and M1$_{global}$ isolates from invasive infections. Percentage of strains with non-synonymous mutations, deletions/insertions, or an inactive protein in 1552 M1$_{UK}$ and 189 M1$_{global}$ isolates is shown. Mutation types are indicated by coloured bars. Percentage of strains where sequence quality precluded analysis (sequence identity <90%) are in dark grey. Differences in *covR* and *covS* mutation frequency between M1$_{global}$ (covR 15/189; *covS* 34/189) and M1$_{UK}$ (*covR* 38/1552; *covS* 106/1552) are significant (one-tailed proportion test: *covR* p < 0.001; *covS* p < 0.001). Ten M1$_{global}$ isolates formed a previously unrecognised clade with covRS mutations. If all strains from this cluster are removed, the *covS* mutation frequency within M1$_{global}$ (24/179) remains significantly greater than M1$_{UK}$ (106/1552) strains (one-tailed proportion test covS p < 0.001). Source data are provided as a Source Data file Fig. 3.

no genomic feature(s) associated only with M1$_{UK}$ from 2022/2023 or M1$_{global}$ from 2022/2023, or the three M1$_{UK}$ clades previously described. Most strains had three prophages typical of *emm*1: Φ5005.1, which encodes *speA*; Φ5005.2, which encodes *spd3* or *spd4*; and Φ5005.3, which encodes another DNase, *sdaD2/sda1*, reported to contribute virulence to modern M1$_{global}$ strains[8]. Although M1$_{UK}$ strains are characterised by increased SpeA expression, 9/1552 (0.6%) invasive M1$_{UK}$ strains had a partial deletion of phage Φ5005.1 including *speA* (Supplementary Table 5). Furthermore, 43/1552 (2.8%) invasive M1$_{UK}$ strains had lost Φ5005.3 and consequently cannot express *sdaD2/sda1*. Prophage Φ370.1 containing *speC* and *spd1* was present in ~10% (174/1815) of *emm*1 strains, 9% (139/1552) in M1$_{UK}$, and 16% (31/189) in M1$_{global}$. Only 4/1815 *emm*1 strains (one M1$_{global}$ and three M1$_{UK}$) from 2014-2020 had the ΦSP1380.vir phage (with *speC, ssa, spd1*) reported in Australia[13] and Hong Kong[18].

*Emm*1 invasiveness has been associated with regulatory gene mutations in vivo[7]. Among iGAS clinical isolates from invasive infection, mutations in the two-component regulatory genes *covR* and *covS* were significantly more frequent in M1$_{global}$ (7.9% and 18% respectively) than M1$_{UK}$ (2.4% and 6.8%) (one-tailed proportion test: *covR p*-value 0.001; *covS p*-value < 0.001) (Fig. 3B), pointing to greater selection pressure on M1$_{global}$ strains during invasive infection. Though observed in both sterile and non-sterile site isolates from invasive infections, this difference in the frequency of *covS* mutations could not be replicated by in vivo passage of non-invasive M1$_{global}$ and M1$_{UK}$ isolates in mice, although only five strains from each lineage were tested using intramuscular inoculation (Supplementary Fig. 3). Mutations in *rgg*1 and *rgg*4 were frequent, but not different between lineages (Supplementary Fig. 4). The frequency of resistance to common antimicrobials among *emm*1 isolates was low (< 0.5%); furthermore,

*pbp2x* missense mutations (T553K and P601L)[19,20] were absent in our dataset (Supplementary Data 2).

**Relationship between non-invasive and invasive *emm*1 isolates**
To extend our analysis to include the reservoir of non-invasive pharyngitis *S. pyogenes* isolates, we sequenced 133 *emm*1 strains collected sequentially from pharyngitis cases in west London in 2022-2023. 14.3% (19/133) of *emm*1 throat isolates were M1$_{global}$ while 85.7% were either M1$_{UK}$ (111/133) or M1$_{UK}$ without the phi5005.3 phage (3/133). Interestingly, the proportion of non-invasive and invasive M1$_{global}$ isolates was higher in London than observed nationally during the same period. Phylogenetic analysis of invasive and non-invasive isolates showed that non-invasive M1$_{UK}$ isolates from west London clustered mostly within Clade 3 (62/111, 55.9%), with other isolates scattered throughout the wider M1$_{UK}$ population, including Clade 1 (8/111, 7.2%) and Clade 2 (4/111, 3.6%) (Fig. 4). The average number of mutations between two isolates from the same clade (Clade 1, 2 or 3) was 2-3 SNPs. 48% (64/133) of non-invasive isolates were found to be identical to at least one invasive isolate (0 SNPs apart, Fig. 4). Point mutations in bacterial regulatory genes in non-invasive *emm*1 sore throat isolates were rare (< 5%), in comparison to invasive isolates. 5/133 (4%) of non-invasive isolates collected in London in 2022 had the ΦSP1380.vir phage.

**Time and place of emergence of M1$_{UK}$ and intermediate lineages**
To elucidate the origin and time of emergence of the M1$_{UK}$ lineage, a dated phylogenetic tree was constructed using a newly sequenced M1$_{UK}$ reference strain H1490 (NCTC14935). The tree comprised 2364 M1$_{UK}$ and intermediate (M1$_{13SNPs}$, M1$_{23SNPs}$ and M1$_{26SNPs}$) genomes collected from Europe (Denmark[21], Iceland[21], Netherlands[22]), United

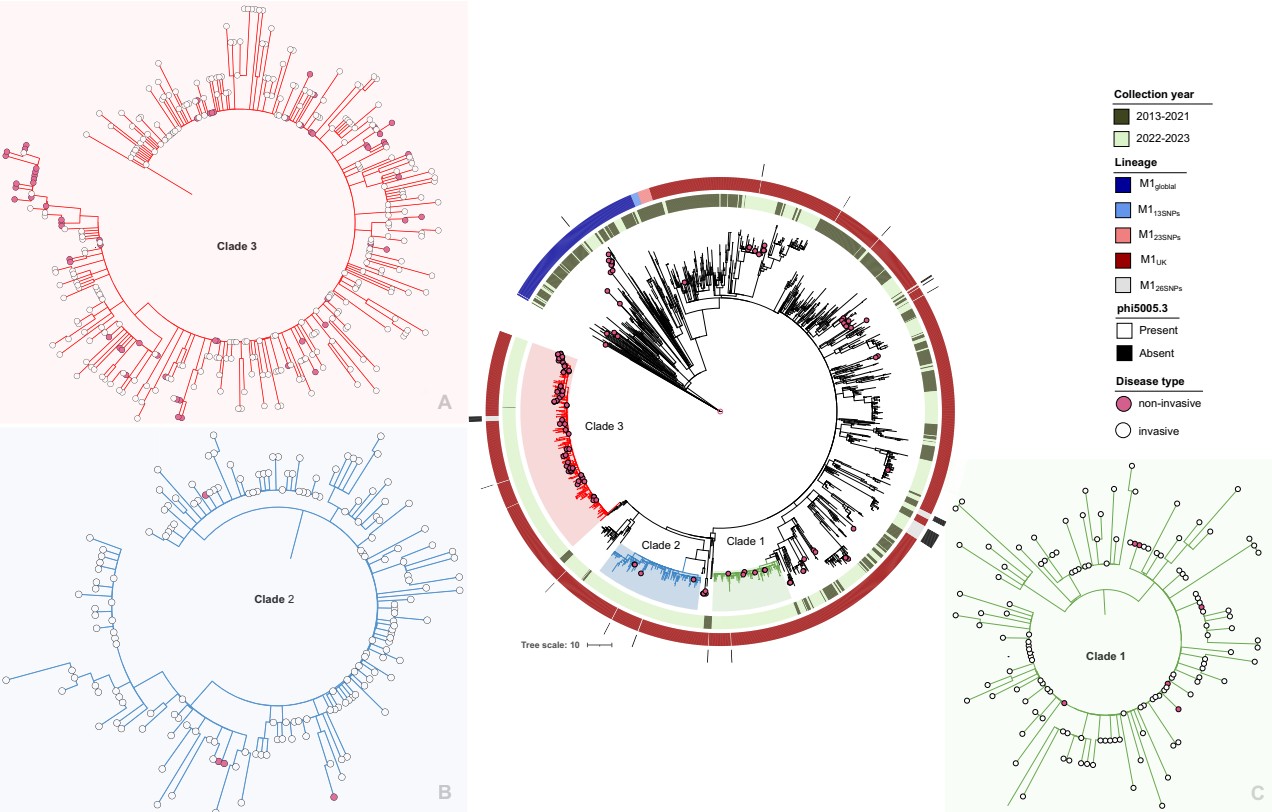

**Fig. 4 | *emm1* phylogenetic tree showing non-invasive sore throat isolates collected in London in 2022 with isolates from invasive infection from UK 2013–2023.** Maximum likelihood phylogenetic tree constructed with the core alignment of 274 SNPs extracted after mapping 1815 *emm1* invasive isolates and 133 non-invasive isolates against MGAS5005. The relationship between invasive and non-invasive infection isolates within Clades1-3 is shown in inset (**A**) Clade 3; (**B**) Clade 2; and (**C**) Clade 1. Source data are provided as a Source Data file Fig. 4.

Kingdom[5,23], plus the isolates from the current study, North America (Canada[24] and USA[25]), and Australia[13] between March 2005 and July 2023. This showed M1$_{13SNPs}$ and M1$_{23SNPs}$ to share a common ancestor with the M1$_{UK}$ lineage, while M1$_{26SNPs}$ are derivatives of M1$_{UK}$ that have lost the Φ5005.3 phage (Fig. 5A). According to the inferred ancestral dates in the tree, the M1$_{13SNPs}$ lineage diverged in 2002 (95% confidence interval (CI): 2000–2004), followed by M1$_{23SNPs}$ in 2006 (95% CI: 2004–2007), and M1$_{UK}$ in 2008 (95% CI: 2006–2009), prior to rapid expansion. The genome-wide mutation rate was estimated to be 1.49 nucleotide substitutions per year.

Ancestral state reconstruction of geographical locations was limited to those regions that undertake and report sequencing of *S. pyogenes*; this revealed that M1$_{UK}$, M1$_{13SNPs}$ and M1$_{23SNPs}$ originated in the UK and then dispersed, with multiple independent introductions into Australia, North America, Netherlands, Iceland, and Denmark (Fig. 5A-B). Denmark and UK strains collected in 2022-2023 were dispersed within the M1$_{UK}$ circulating population, including Clade 3, while almost all 2022-2023 Iceland isolates grouped together in Clade 2.

Bayesian inference of the M1$_{UK}$ effective population size through time in the UK demonstrated rapid population growth of M1$_{UK}$ from 2008 until 2015, followed by a progressive decline until 2019, and then a sharp decline in early 2020 (Fig. 5C). Strikingly, the population dynamics suggested a transmission bottleneck in M1$_{UK}$ during the implementation of severe NPI designed to limit the spread of COVID-19 (April 2020 –March 2021). The mean effective population size over this period dropped to one-fifth of the pre-pandemic maximum and then rose steeply after the lifting of the lockdown and other NPI measures. Importantly the inferred patterns of population growth and decline were not driven by any variation in the number of sequenced M1$_{UK}$ isolates in the UK through time (Supplementary Fig. 5).

## Discussion

The marked increase in bacteriologically confirmed *S. pyogenes* infections in England in late 2022-2023 coincided with the reported national upsurge in notifications of both scarlet fever and iGAS[15,16]. The upsurge in invasive infections was clearly associated with a significant increase in *emm*1 *S. pyogenes* only, the vast majority (95.7%) of which belonged to the emergent M1$_{UK}$ lineage or its derivatives. No substantial genomic changes in M1$_{UK}$ were observed during the upsurge, but three new clades emerged and expanded within M1$_{UK}$, accounting for 53% of *emm*1 iGAS in 2022-2023.

Several countries have now reported similar iGAS upsurges in the period 2022-2023, chronologically associated with the end of mitigation strategies implemented during the COVID-19 pandemic[21,26–29], including association with *emm*1[26] or M1$_{UK}$[28,29]. Although *emm*12 infections were prominent in early 2022[15] as reported elsewhere[29,30], the very marked increase in iGAS observed in the second half of 2022 in England was accounted for by *emm*1. The dominance of *emm*1 among invasive isolates ( > 50% overall, and almost 70% in children) is unprecedented in UK records. In contrast, during the period of COVID-19 pandemic-related NPI in 2020-2022, bacteriologically confirmed *S. pyogenes* infections were rare. While reduction in non-invasive infection detection might be explained by a reduction in consultations, this would not explain the reduction in sterile site isolates. Furthermore, during the period of COVID-19 NPI, invasive infections due to *emm*1 were exceedingly rare, with no *emm*1 isolates identified during some quarters of 2020-2021 in either adults or children. We posit this points to differential modes of transmission, whereby 'throat specialist' strains[31] such as *emm*1 require respiratory transmission in order to circulate, while others may spread via skin contact.

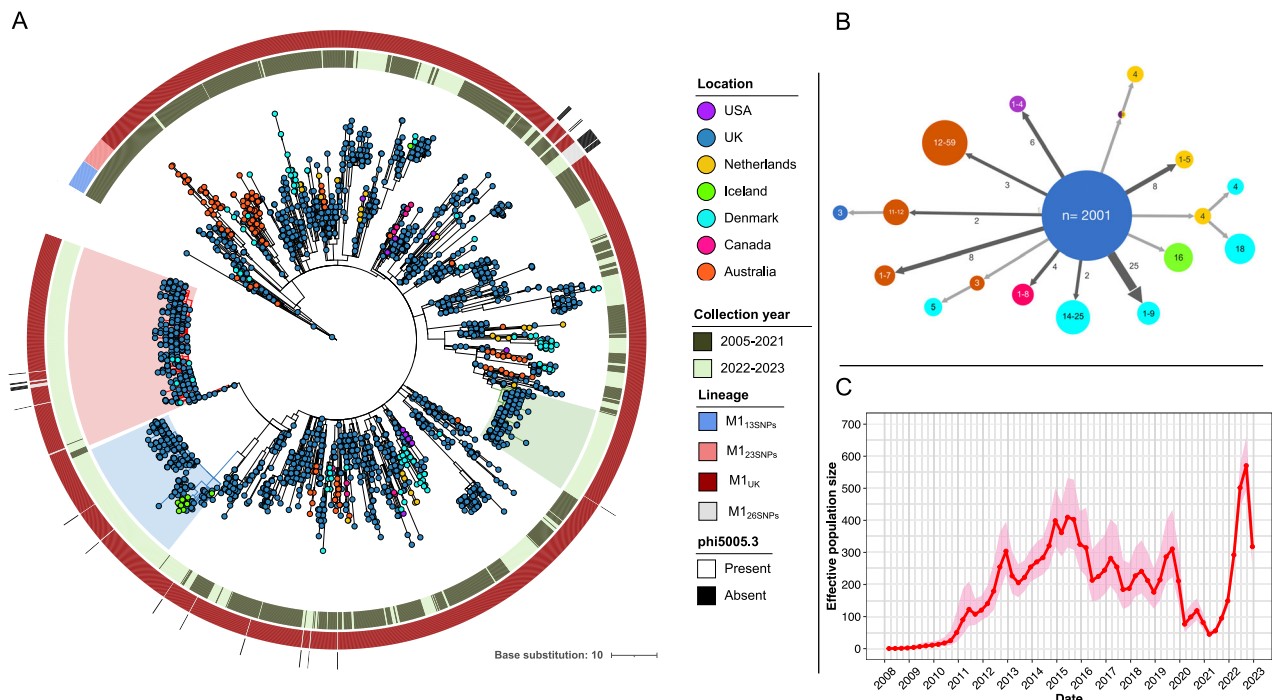

**Fig. 5 | Global distribution and potential introduction events of M1_UK and intermediate populations. A** Phylogenetic tree of 2364 M1_UK and intermediate strains collected globally March 2005 to July 2023. The tree was built based on 3406 SNPs from a core genome alignment relative to M1_UK (H1490/NCTC14935) reference genome and rooted on a closely related M1_global genome gas81595 (also included in this tree). Leaves are coloured based on the country where samples were collected. Shading indicates the 3 emergent clades (Clade 1, green; Clade 2, blue; Clade 3, red). Coloured bars in concentric circles represent (from inside to outside): collection years (pre-upsurge 2013-2021 and upsurge 2022-2023); *emm*1 lineage; and presence/absence of the phi5005.3 phage. **B** Simplified transmission tree by PastML showing the ancestral epidemic location of M1_UK and intermediate lineages. Each node represents a cluster of leaves sharing the same probable ancestral location and is labelled by the range of leaves numbers. Each arrow indicates inferred international transmission events; arrow width and labels indicate the number of identical origin-destination transmission events. For example, the arrow labelled "6" pointing at the node "1–4" (USA) indicates six clusters of 1 to 4 leaves were present in the USA that were likely imported from the UK. **C** Estimated effective population size ($N_e$) of M1_UK in the UK through time. The red line and pink shading at each time point indicate the mean and 95% confidence interval of $N_e$, respectively. Source data are provided as a Source Data file Fig. 5.

The reported increase in iGAS in late 2022 was particularly evident in children, with complicated clinical presentations including meningitis[28] and, specifically, rapidly progressive pleural empyema in countries where such data are collected[15,29]. Isolates from empyema are often not cultured due to antibiotic pre-treatment. Hence, the pleural sample isolates in the current study represent a subset of all pleural empyema cases. Regardless, pleural isolates were significantly associated with the emergent M1_UK clades. The timing of the upsurge in Q3 2022 is very likely to have contributed to the pleural empyema phenotype; respiratory viral infections were identified in 25% of paediatric cases of empyema[15], playing a potential role in progression to lower respiratory tract infection. Due to the design of our study and the widespread adoption of respiratory viral point-of-care tests to diagnose respiratory viral infection in 2022, we are unable to assess the effect of respiratory viral infection as a contributor to empyema over time in the current study.

M1_UK is increasingly dominant in the UK. Our findings are mirrored to different degrees in other countries, where the proportion of *emm*1 isolates that are M1_UK ranges from 41.5%-78%[21,28,29]. The fitness of M1_UK has been attributed to its ability to express SpeA, a superantigen that can promote pharyngeal infection[5]. Increased SpeA is associated with a SNP in the leader sequence of *ssrA*[13], which is present in not only M1_UK but also the near-extinct intermediate M1_23SNPs lineage. The contraction of the M1_23SNPs lineage suggests that additional fitness advantages prevail in M1_UK[6]. Genome stability appeared greater in M1_UK than M1_global, suggesting the accumulated 27 SNPs in M1_UK may be sufficient to confer a fitness advantage during human infection, including increased transmissibility. Indeed, in one study, the mean

secondary attack rate was 40% among asymptomatic contacts of M1_UK infection in two classes of schoolchildren, compared with 22.8% in classroom outbreaks involving different *emm* types[32], supporting a potential transmission advantage. In the current study, M1_UK invasive isolates were significantly less likely to exhibit mutations in *covRS* than M1_global strains, suggesting a fitness advantage in invasive infection as well. Although we were unable to reproduce this difference experimentally, the intramuscular route of infection in mice does not reflect the bottleneck of natural mucosal infection in humans and was necessarily limited to just five strains per group.

A comparison of non-invasive *emm*1 isolates from London and invasive *emm*1 isolates nationally revealed both groups to be interspersed and clustered tightly in the phylogenetic tree, indicating a common genetic pool. The analysis showed that individual invasive isolates can be derived repeatedly from the population of pharyngitis strains. The identical nature of strains underlines the route of direct transmission from cases of pharyngitis and scarlet fever to dangerous invasive infections, often unnoticed. We found that diversifying selection in the invasive population, especially in M1_global, drives the accumulation of mutations in *covRS*, as reported[33].

Our study evaluated the origin, dispersion, and population dynamics of M1_UK by assembling the most comprehensive global collection of M1_UK strains to date. The analysis showed M1_UK to be globally distributed, with nearly identical strains found all over the world and multiple introductions from the UK population. The 2022 upsurge in the UK was characterised by the rapid expansion of three clades within M1_UK, of which two showed swift dispersal to at least two other European countries. In Iceland, a single introduction event appeared

responsible for reported M1$_{UK}$ cases, whereas in Denmark, multiple introductions seemed likely. We found no evidence of importation of a new lineage recently reported in Denmark (M1$_{DK}$)[21].

The origin of M1$_{UK}$ was estimated to date from 2008, the year in which national guidelines to reduce swab testing and unnecessary antibiotic treatment of sore throat were introduced in England[10]. An exponential increase in the M1$_{UK}$ population commenced around 2010. Given the propensity for M1$_{UK}$ to spread readily in classrooms[32], it is conceivable that new lineages can emerge and rapidly expand if active *S. pyogenes* throat infections are not detected and treated with antibiotics and transmission is not controlled. Antecedent intermediate lineages emerged in 2002 (M1$_{13SNPs}$) and 2006 (M1$_{23SNPs}$), during which time secular changes in sore throat management were ongoing in the UK[34,35].

Our dataset is limited to the UK and other high-income temperate countries, hence no inferences about the importation of M1$_{UK}$ into low-income countries were possible. This underlines the importance of global surveillance to monitor the evolution and epidemiology of emerging variants with increased capacity for pathogenicity. Although M1$_{UK}$ geographic origin was identified as the UK, this was the only country with genomes available from the time of emergence, as such, we cannot exclude an alternate origin.

The phylodynamic analysis of M1$_{UK}$ in the UK showed a decline in population size between 2015-2019 after the initial rapid rise, consistent with the cyclical changes in *S. pyogenes* populations known to occur[36], however population size plummeted in early 2020 when NPI to combat spread of COVID-19 were introduced. The marked M1$_{UK}$ population bottleneck was followed by rapid expansion in 2022 and 2023, raising the question of whether strain-specific survival advantages exist during periods of such low *S. pyogenes* population activity. Global reductions in other bacterial respiratory pathogens were seen during the period of COVID-19 NPI[37]. However, the scale of resurgence in invasive *S. pyogenes* following the relaxation of NPI thus far appears unique, perhaps related to the lack of a vaccine for *S. pyogenes* compared with other pathogens studied[37]. The observed magnitude and severity of the upsurge could be explained by the coincidence of enhanced M1$_{UK}$ pathogenicity and diminished human population immunity to *S. pyogenes*, as a predictable but perhaps unintended consequence of interventions to limit the spread of COVID-19[38]. The role of exposure-driven human immunity in shaping cyclical and post-COVID-19 changes in *S. pyogenes* epidemiology is the subject of ongoing research. Scarlet fever affects children in their first year of school[39], an experience that was delayed for many during two years of COVID-19-related NPI. We hypothesise this resulted in a ~3-fold increase in susceptible children starting school in Q3 2022, with a similar reduction in immunity in siblings and adults. We posit that the transmissibility and invasiveness of M1$_{UK}$ facilitated the exponential and unprecedented increase in invasive *S. pyogenes* infections.

## Methods

### Surveillance of *S. pyogenes* detection in clinical samples in England

UK Health Security Agency surveillance of infections for health protection purposes is approved under Regulation 3 of The Health Service (Control of Patient Information) Regulations 2020 and under Section 251 of the NHS Act 2006. All reports of *S. pyogenes*-positive clinical samples, including post-mortem, from ISO-8601 week 1 2016 to week 30 2023 reported by English laboratories were extracted from the UK Health Security Agency (UKHSA) Second Generation Surveillance System (SGSS) on 7 December 2023. SGSS captures approximately 98% of electronically supplied hospital microbiology laboratory data in England; however, is the primary route for statutory reporting[40] of laboratory-confirmed invasive *S. pyogenes* infections. Invasive *S. pyogenes* samples are defined as culture-positive samples (or positive by

molecular detection) obtained from a normally sterile site. *S. pyogenes*-positive samples were deduplicated where patients had more than one positive *S. pyogenes* similar specimen type taken on the same date.

### Invasive Streptococcus pyogenes isolates

*S. pyogenes* isolates from invasive disease (iGAS) cases in England, Wales, and Northern Ireland are routinely submitted to the national reference laboratory (SSRS, Staph and Strep Reference Section, UKHSA, London, UK) for *emm* genotyping using standard methods (https://www.cdc.gov/streplab/groupa-strep/emm-typing-protocol. html). Processes and reporting requirements for isolate submission, including clinical sample source, were unchanged during the study period. The percentage of invasive isolates that were determined to be *emm*1 was determined compared with the overall total number of isolates genotyped. As part of the investigation into the upsurge of *S. pyogenes*, all *S. pyogenes* isolates from 2022/23 were whole genome sequenced (WGS). For this study, we included all *emm*1 isolates from invasive infections that had been genome sequenced at the reference laboratory from 2014–2023, including a small number from other regions. This included *emm*1 isolates from 2014-2015, previously reported ($n = 516$)[41]; *emm*1 isolates from 2016–2021 ($n = 207$) intermittently sequenced as part of service delivery; and all *emm*1 strains ($n = 1092$) submitted to the reference laboratory from January 2022-March 2023 that were sequenced as part of this outbreak investigation. Metadata and accessions for all isolate genome sequences are listed in Supplementary Data 3. Isolate WGS was linked to reported clinical sample type. Differences in the proportion of *emm*1 between time points were evaluated using a one-tailed proportion test (https://www. socscistatistics.com/tests/ztest/).

### Non-invasive *S. pyogenes* isolates

The collection and analysis at Imperial College London of fully anonymised bacterial isolates from a diagnostic laboratory previously linked to routine data was approved by a national research ethics committee (West London Research Ethics Committee 06/ Q0406/20). *S. pyogenes* throat isolates were identified by MALDI-Biotyper (Bruker) from swabs submitted to the Diagnostic Laboratory at Imperial College Healthcare NHS Trust (London, UK) during 2022 (1 January - 31 December). This laboratory serves northwest London, a population of ~2 million people, representing ~3.5% of the population of England. *S. pyogenes* isolates were cultured on Columbia Blood Agar (CBA, Oxoid, Basingstoke, UK) or in Todd Hewitt broth (Oxoid) at 37 °C with 5% $CO_2$. Demographic data were linked to all isolates and anonymised in accordance with the approved protocol (06/Q0406/20). All *emm*1 pharyngitis isolates (from throat swabs) were genome sequenced at the National Reference Laboratory (Supplementary data 3).

### Genomic data contextualisation

Three different genomic datasets were included in this study. The first contains 1815 (1092 newly sequenced from 2022-2023; and 723 from 2013-2021) *emm*1 strains associated with invasive infections collected at the national level and sequenced at the UKHSA national reference laboratory from 2013 to 2023 (Supplementary Data 3); 12 isolates were from outbreak investigations. The second dataset contained the 1815 invasive strains described above plus 133 newly sequenced non-invasive *emm*1 isolate whole genome sequences (WGS) collected in London during 2022 as part of this study (1 January to 31 December), yielding a total of 1948 *S. pyogenes* isolate WGS (Supplementary Data 3). The third dataset was created to provide phylogenetic context for the M1$_{UK}$ global population and intermediate strains only. This dataset included an additional 385 previously-published M1$_{UK}$ or intermediate WGS from the UK sequenced at the Wellcome Trust Sanger Institute dating from 2005-2018[5,23]; 163 M1$_{UK}$ or intermediate

WGS collected in Australia 2010-2022[13]; 16 M1$_{UK}$ or intermediate WGS collected in Canada 2016-2019[24]; 120 M1$_{UK}$ or intermediate WGS collected in Denmark 2018-2023[21]; 18 M1$_{UK}$ or intermediate WGS from Iceland, 2023[21]; 27 M1$_{UK}$ or intermediate WGS collected in the Netherlands 2019[22]; and 10 M1$_{UK}$ or intermediate WGS from USA collected in 2015-2018[25]. Data collection finished in July 2023, and therefore genomes reported after that time point were not included. The final global dataset contained 2365 M1$_{UK}$ and intermediate strains (Supplementary Data 3).

### Generation of new M1$_{UK}$ reference genome: Reference strain NCTC14935

Genomic DNA from *S. pyogenes* M1$_{UK}$ isolate H1490 and M1$_{global}$ isolate H1499 (both sore throat isolates) was sheared using a Megaruptor to prepare 20-22 kb PacBio SMRT libraries, following the manufacturer´s recommendations. The libraries were sequenced using one Single Molecule Real-Time (SMRT) cell in a PacBio RSII platform (Pacific Biosciences of California, Inc., Menlo Park, CA, USA) at the University of Edinburgh. The data was demultiplexed using Lima v2.2.0 (https://lima.how/). The demultiplexed CLR data was converted to CCS using ccs tool v6.3.0, and further HiFi reads (CCS > Q20) were extracted using extract hi fi tool from the same package. The genome assemblies were generated from the HiFi reads using Redbean v 2.25[42] and Trycycler v0.5.3[43]. The assembly quality was assessed using QUAST v5.0.2[44] and BUSCO v5.3.0[45]. The annotation was performed using prokka v1.14.6[46]. PacBio sequencing reads and data are deposited in the European Nucleotide Archive under BioProject accession PRJEB68198 (M1$_{UK,}$ H1490 - ERR12378139 and M1$_{global,}$ H1499 - ERR12378140). The two isolates have been deposited in the National Collection of Type Cultures (NCTC) with the accessions NCTC14935 (M1$_{UK}$, H1490) and NCTC14936 (M1$_{global}$, H1499).

### Illumina genome sequencing, assembly, and annotation

For this study, whole genome sequencing of all clinical isolates (invasive and non-invasive) was performed by the UKHSA reference laboratory using the Illumina NextSeq 1000 platform with 100 base paired-end chemistry. Reads were trimmed to remove adaptor sequences and low-quality bases with Trimmomatic v0.39[47]. Contamination was assessed based on Kraken2[48] classification of reads mapped against a standard database for bacteria. Genomes with less than 90% of the reads mapped against *S. pyogenes* were excluded. Draught genomes were generated using SPAdes v3.15.4[49]. The assembly quality was assessed using QUAST v5.0.2[44], and poor assemblies were filtered out if the genome size was higher than 2.1 Mbp and/or had more than 400 contigs. Genome annotation was performed with prokka v1.14.6[46].

### Identification of single nucleotide variations and phylogenetic analysis

Core genome alignment was obtained by mapping trimmed reads of *S. pyogenes* genomes to MGAS5005 (GenBank accession: CP000017.2) reference genome using snippy v4.6.0 (https://github.com/tseemann/snippy), with a minimum coverage of 10, a minimum fraction of 0.9 and minimum vcf variant call quality of 100. The SNP distance matrix was obtained using snp-dist (https://github.com/tseemann/snp-dists). SNPs identified were classified as non-coding, missense or synonymous according to their location in the genome and their effect on protein sequence using Snippy. Gubbins v3.3.0[50] was used to identify and remove recombinant regions. A maximum-likelihood (ML) phylogenetic tree was constructed from the multi-sequence alignment using RAxML-NG v1.0.1[51] implemented in Gubbins v3.3.0 (substitution and rate variation model: GTR + Gamma). The ML tree was rooted on NCTC8198 (GenBank accession: GCA_002055535.1, reference genome of old *emm1* lineage). Phylogenetic trees and associated data were visualised using iTOLv6.8.1[52].

### Characterisation of genomic features of interest

The presence of AMR genes was predicted by combining the results from ABRicate (https://github.com/tseemann/abricate), Ariba[53] and srst2[54]. The *pbp* gene sequences (*pbp1a*, *pbp1b*, and *pbp2x*) were obtained using a BLASTN (NCBI BLAST+ v2.7.1) search. The nucleotide sequences were converted to amino acids and examined for the presence of non-synonymous mutations. None of the non-synonymous mutations previously associated with penicillin resistance in *S. pyogenes* were identified. A similar approach was used to identify non-synonymous mutations in *S. pyogenes* regulatory genes (*covR, covS, fasA, fasB, fasC, rgg1, rgg2, rgg3, rgg4, rivR, rofA* and *rocA*). The presence of superantigens (*smeZ, speA2, speC, speG, speH, speI, speJ, speK, speL, speM, speN, speO, speP, speQ, speR, ssa*) and DNAses (*sda2, sdn1, spdn1, spd3, spd4, spdB, spnA*) was accessed with a BLASTN (NCBI BLAST+ v2.7.1) analysis with the default parameters. Differences between lineages (M1$_{global}$ and M1$_{UK}$) regarding the number/type of mutations found in regulatory genes and *pbp* genes were evaluated using a one-tailed proportion test (https://www.socscistatistics.com/tests/ztest/). Regulatory gene sequences with <90% similarity to the reference genome were excluded from the identification of regulatory gene mutations.

### Pangenome analysis

A pangenome graph was constructed from annotated genome assemblies of MGAS5005 and 1815 *emm1* isolates collected from across the UK between 2013 and 2023 using Panaroo v1.3.0[55] under its moderate decontamination mode. Clusters of orthologous genes (COGs) were defined by a minimum nucleotide identity of 98%, and core genes were defined by a minimum frequency of 95%. The resulting gene presence-absence matrix was filtered to remove pseudo and fragmented genes as well as those of unusual lengths. The pangenome graph was simplified with the MGAS5005 genome as a reference using Panaroo's helper script *reference_based_layout.py* for visualisation in Cytoscape v3.10.1[56]. Presence-absence of COGs was compared between M1$_{UK}$ and M1$_{global}$ and between pre-2022/2023 and 2022/2023 groups using Python v 3.11.6.

### Phylodynamic analysis of M1$_{UK}$

A maximum-likelihood (ML) phylogenetic tree corrected for recombination events was constructed from the multi-sequence alignment of global M1$_{UK}$ and intermediate genomes (against the M1$_{UK}$ reference genome H1490) using RAxML-NG v1.0.1[51] as implemented in Gubbins v3.3.0[50] (model: GTR + Gamma). The ML tree was rooted on M1$_{global}$ isolate gas81595 (ERS17508611), which was the most closely related to M1$_{UK}$ and intermediate lineages according to SNP distances. A dated phylogenetic tree was generated from the ML tree using the least-squares dating method implemented in the LSD2 module of IQ-Tree v2.2.2.7 (model: GTR + I + G4)[57,58]. Ancestral geographical locations were inferred from the dated tree and isolate information using the MPPA method and F81 model as implemented in PastML[59].

To reconstruct the population dynamics of the M1$_{UK}$ lineage in the UK, a UK-specific subtree of M1$_{UK}$ genomes was extracted from the dated tree, and the M1$_{UK}$ effective population size (N$_e$) was thereby modelled through time using a skygrowth model[60] implemented in R package mlesky (with 60-time intervals as determined using the package's parameter-optimisation algorithm based on the Akaike Information Criterion)[61]. Furthermore, the same model was iteratively fitted on 40 subtrees of randomly sampled UK M1$_{UK}$ genomes (with a maximum of 76, 22, and 14 genomes per year, respectively, based on sample sizes between 2019 and 2021) to evaluate if the variation in sample size over time could impact the inference of N$_e$.

### In vivo screening for *covR/S* mutations using five representative strains of M1$_{UK}$ and M1$_{global}$

All animal experiments were undertaken using protocols approved by the Imperial College Animal Welfare Advisory Board (AWERB) and

authorised by a UK Home Office Project Licence. Mice were maintained in a standard 12 h light/12 h dark cycle with food and water available ad libitum.

Five $M1_{global}$ and five $M1_{UK}$ strains were used in this study (Supplementary Fig. 3). Strains were selected from isolates from 2022 that were broadly representative of each lineage and that had no existing covR or covS mutations. Experimental soft tissue infections were performed using female BALB/c mice aged 6 weeks (Charles River, UK). Bacteria were cultured on CBA overnight and resuspended in sterile PBS. Mice were infected with $5 \times 10^8$ CFU of one of the 10 strains (3 mice per strain) into the thigh muscle. 24 h after infection, mice were sacrificed, and 150 μl heparinized blood obtained by cardiac puncture from each mouse was plated onto CBA prior to euthanisation. Each spleen was removed, homogenised using FastPrep-24™ 5 G in 1 ml PBS and plated on CBA for enumeration. Agar-based casein digestion assay was used to determine SpeB activity to infer covS mutations. 50 colonies cultured on CBA from spleens were patched onto 2% w/v skim milk Todd Hewitt agar (THA) to determine SpeB activity. One spleen sample with only a single colony was excluded from analysis; three samples with 16, 33 and 36 colonies were included. Fifty colonies from the inoculum of each strain were patched onto skim milk THA to rule out covS mutations occurring before introduction to the mice. SpeB (caseinolytic) activity was determined by comparing zones of clearance from S. pyogenes isolates to positive controls on the same plates and repatched to confirm the phenotype. Statistical analysis was performed with GraphPad Prism 10. A comparison of the two groups was carried out using a two-tailed nested t-test.

### Reporting summary

Further information on research design is available in the Nature Portfolio Reporting Summary linked to this article.

## Data availability

The genome sequences (PacBio sequences) of $M1_{UK}$ and $M1_{global}$ reference strains generated in this study have been deposited in the ENA database under the BioProject PRJEB68198. Illumina short reads of all 1815 emm1 S. pyogenes used in this study from invasive disease cases (from the UK, 2013 to 2023) were deposited under the BioProject PRJEB68199. Illumina short reads of emm1 non-invasive disease pharyngitis isolates collected in London in 2022 were deposited under the BioProject PRJEB71329. Metadata relating to newly sequenced UK isolates and other genome sequences used in this work are provided in Supplementary Data 3 and in associated source data files. Detailed demographic information is protected due to privacy rules. Genome assemblies and metadata of 2365 $M1_{UK}$ isolates analysed in this study are available as a collection on Pathogen Watch (https://pathogen.watch/collection/6pssdapzoqg5-m1uk-and-intermediates-vieira-et-al-2024). Reference genome MGAS5005 is available at https://www.ncbi.nlm.nih.gov/datasets/gene/GCA_000011765.2. Source data are provided in this paper.

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

## Acknowledgements

The authors are grateful to the local diagnostic laboratories and microbiologists who submit isolates to the reference laboratory for genotyping and to the Imperial College Healthcare Trust diagnostic laboratory for bioresourcing  throat isolates for this study. The authors are also grateful to laboratories around the world that publish and share *S. pyogenes* genome sequences. SS acknowledges support from the NIHR Imperial Biomedical Research Centre, which also supports the BRC Leonard and Dora Colebrook Laboratory. HKL is a Medical Research Council (MRC) CMBI Clinical Training Fellow; VWCS is an MRC Career

Development Fellow; EJ is an Imperial College Research Fellow, funded by Rosetrees Trust and the Stoneygate Trust; NJC and EJ acknowledge the MRC Centre for Global Infectious Disease Analysis funded by the MRC and Department for International Development (grants MR/R015600/1 and MR/T016434/1). AV, YW, EJ, JC, TL, CSB, and SS are affiliated with the NIHR Health Protection Research Unit in Healthcare Associated Infections and Antimicrobial Resistance at Imperial College London in partnership with the UK Health Security Agency (formerly Public Health England), in collaboration with Imperial Healthcare Partners, University of Cambridge and University of Warwick; XD is affiliated with the NIHR Health Protection Research Unit in Genomics and Enabling Data at the University of Warwick. The views expressed in this publication are those of the author(s) and not necessarily those of the NHS, the National Institute for Health Research, the Department of Health and Social Care or the UK Health Security Agency. This work was supported by grants from the UK Medical Research Council to SS (MR/P022669/1 and MR/X001962/1); with additional support from the UK National Institute for Health Research (NIHR) Imperial College Biomedical Research Centre (BRC); NIHR Health Protection Research Unit in Healthcare Associated Infections and Antimicrobial Resistance at Imperial College London; the Conor Kerin Foundation; Illumina sequencing was funded by the UK Health Security Agency.

## Author contributions

Conceived and designed the study: S.S. and J.C. Data analysis: A.V., Y.W., Y.R. and V.W.C.S. Data collection: H.K.L., M.P., K.K.H., L.C.R., R.D., A.H., K.B., C.D., M.G., M.A.G., Z.M., M.R., E.R., R.M., E.A.M., V.S. and K.Y.M. Data visualisation: A.V., Y.W., R.L.G. Data analysis tools: X.D. and N.J.C. Project supervision S.S., J.C., T.L. and C.S.B. Analysis supervision: E.J., X.D., N.J.C. Writing, first draft: A.V., Y.W. and S.S. Editing and approval of the final manuscript: All.

## Competing interests

The authors declare no competing interests.
