## [Peer Review File · Nature Communications]

REVIEWER COMMENTS

Reviewer #1 (Remarks to the Author):

Manuscript title: Rapid expansion and international spread of M1UK in the post-pandemic upsurge of *Streptococcus pyogenes* infections in United Kingdom.

Manuscript#: NCOMMS-24-02520-T

This is a manuscript describing the surge in group A streptococcal infections in the United Kingdom. The authors have used whole genome sequencing to explore the diversity or lack in the emm1 population of group A streptococcal isolates collected from years 2013 to 2023. In their report, the authors focus primarily on the years associated with the upsurge in cases in the UK 2022 to 2023.

The authors characterize circulating M1UK strains and identify three clades that recently have arisen. Data is also provided regarding the snps associated with each clade. In their analysis, the authors present data indicating that M1UK likely arose in 2008 in the UK and underwent an expansion followed by a bottleneck and then rapid expansion after covid restrictions were lifted. Authors also present data regarding the association of M1UK isolates with pleural empyema during the upsurge in the UK.

The manuscript is well written and provides novel, important and topical information regarding the surge in group A streptococcal infections being seen throughout the northern hemisphere and not just the UK. The authors have presented a large amount of work regarding the M1UK strains associate with upsurge through the use of whole genome sequencing and analysis of these sequences. I have very few comments for the authors.

Minor comments for the authors:

1. line 98 to 99. Does the “national reference laboratory” have a name? This is the first it is mentioned in the manuscript. I have looked in the materials and methods section and also do not see the name. It may have been indicated elsewhere in the manuscript and I have missed it.
2. Line 120. “...the lineage since 2020 (Figure 3A).” Is it 2020 or 2022. Paragraph starts with discussion of data from 2022 to 2023.
3. Lines 371 to 373. I assume these were throat swabs only and not skin or wound swabs?

Reviewer #2 (Remarks to the Author):

The manuscript by Vieira et al. describes the rapid expansion and spread of an M1-UK lineage of group A Streptococcus (GAS) during the post-pandemic period. The study is founded on the observation that in late 2022 and early 2023, the UK observed a rapid, exponential rise in the number of non-invasive and invasive GAS infections (national surveillance). The peak was notable given the relative absence of GAS infections (non-invasive and invasive) during the peak pandemic period and for being substantially higher than previous, seasonal outbreaks in the same region. Compared to previous periods, the post-pandemic surge was noted to be dominated by emm1 GAS. GAS WGS and phylogenetic analysis showed an apparent bottleneck during the pandemic resulting in emergence of 3 dominant clades and several smaller clades during 2022/2023. The dominant clades were clonal in nature with some disease association (pleural samples more common especially in clade 3). Interestingly, frequency of covRS mutation (known to occur with some frequency in invasive GAS strains) in the M1-UK (post-pandemic) was less frequent than described in M1 GAS representative of more global populations (M1-global). Expansion of the genomic analysis to include international M1 isolates (M1-global) demonstrated multiple independent emergence events of the M1-UK strain into different settings globally. The authors conclude that a transmission bottleneck in M1-UK during the pandemic period, presumably from non-pharmaceutical interventions, combined with unresolved host factors was the main driver of the rapid expansion in the post-pandemic period.

Overall, the manuscript is well-written, flows logically, methodologically sound and rigorous, and conclusions supported by the presented data. The study is impactful in that it provides an important clue to the upsurge of GAS infections observed during the post-pandemic years. Factors limiting the novelty and/or impact of findings include (i) the multiple observations (as noted by the authors) of GAS undergoing similar epidemic clonal emergence followed by decline; (ii) that the upsurge of GAS due to M1-UK was limited to the UK (no evidence that similar events due to M1-UK have occurred in other geographic regions); and (iii) lack of a mechanism for the observed increase in M1-UK (posit that due to known features of M1-UK but why then does there seem to be some predilection for pediatric populations and increased frequency from pleural samples)? Thus, the broad interest in the study may be limited.

Specific comments:

1. Results, line 145: “pleural samples compared to [M1-global]” – are the sources/metadata harmonized between the two populations over time? Also, how significant is misclassification bias (i.e., patients with GAS from blood but with pulmonary/pleural disease and no pleural sample) in either dataset? These may impact the comparison made.

2. General: Association with pediatric and pleural samples – while statistically significant, there is generally a lack of mechanism for this association. It is possible, and in this reviewer’s opinion highly likely, that the increased frequency of respiratory viruses during this same time period is a major contributor to the observed association (pediatric AND pleural isolation). Do the authors have data regarding concomitant respiratory viral infection/disease? It seems less likely, based on data presented

on the post-pandemic M1-UK strains, that there is a bacterial genetic/phenotypic factor that contributes to this association but has not been tested.

3. Ext data Figure 3 – No explanation of where these strains are derived from the larger population of M1-global or M1-UK. Of what are they representative?

4. Discussion, lines 254-256: Agree that “throat specialists” and the mode of transmission is/was a critical factor during the pandemic GAS disease decline and post-pandemic surge. This is supported by the data presented by Aboulhossn et al. (PMID 37011014) that observed a similar disproportionate post-pandemic spike in emm12 GAS.

5. Discussion, lines 324-325: “raising the question of how such strains are maintained during periods of low [GAS] activity.” Undoubtedly, this is due to asymptomatic carriage and any number of data/studies are available to support this.

6. Discussion, lines 326-329: This is a known phenomenon called the “divorce effect” (PMID 33075052).

Point by Point Response to Reviewers (Rapid expansion and international spread of M1_{UK} in the post-pandemic upsurge of *Streptococcus pyogenes* infections in United Kingdom: NCOMMS-24-02520-T)

Reviewer #1 (Remarks to the Author):

The manuscript is well written and provides novel, important and topical information regarding the surge in group A streptococcal infections being seen throughout the northern hemisphere and not just the UK. The authors have presented a large amount of work regarding the M1_{UK} strains associate with upsurge through the use of whole genome sequencing and analysis of these sequences. I have very few comments for the authors.

We are grateful to the Reviewer for these supportive comments

Minor comments for the authors:

1. line 98 to 99. Does the "national reference laboratory" have a name? This is the first it is mentioned in the manuscript. I have looked in the materials and methods section and also do not see the name. It may have been indicated elsewhere in the manuscript and I have missed it.

The national reference laboratory is based at the UK Health Security Agency and is known as the SSRS (Staph and Step Reference Section); we now provide this name in the Methods.

2. Line 120. "...the lineage since 2020 (Figure 3A)." Is it 2020 or 2022. Paragraph starts with discussion of data from 2022 to 2023.

We apologise for lack of clarity. We have now re-phrased this as "Taken together, 95.7% of all *emm1* strains from the 2022/2023 upsurge period were M1_{UK} or a derivative thereof, representing overall expansion of the lineage since 2020 (ref 14)"

3. Lines 371 to 373. I assume these were throat swabs only and not skin or wound swabs?

Yes, only throat swab isolates were used in this study; we have now clarified this as throat swabs only, thank you for highlighting this.

Reviewer #2 (Remarks to the Author):

Overall, the manuscript is well-written, flows logically, methodologically sound and rigorous, and conclusions supported by the presented data. The study is impactful in that it provides an important clue to the upsurge of GAS infections observed during the post-pandemic years.

We are encouraged that the Reviewer found the study to be impactful.

Factors limiting the novelty and/or impact of findings include (i) the multiple observations (as noted by the authors) of GAS undergoing similar epidemic clonal emergence followed by decline; (ii) that the upsurge of GAS due to M1-UK was limited to the UK (no evidence that similar events due to M1-UK have occurred in other geographic regions) ; and (iii) lack of a

mechanism for the observed increase in M1-UK (posit that due to known features of M1-UK but why then does there seem to be some predilection for pediatric populations and increased frequency from pleural samples)? Thus, the broad interest in the study may be limited.

- (i) the multiple observations (as noted by the authors) of GAS undergoing similar epidemic clonal emergence followed by decline.

We accept and acknowledge that *S. pyogenes* is known to undergo periodic epidemic clonal emergence and then decline. The scale of M1_{UK} expansion (both in terms of total invasive isolates and in terms of the proportion of iGAS caused) is however unprecedented in the modern era. Emergence of the SLO-producing *emm1* clone (M1_{global}) in the 1980's accounted for a maximum of 28% of bloodstream infections, peaking in 1987. (doi: 10.1099/00222615-39-3-165). In contrast, the proportion of iGAS caused by M1_{UK} appears to be greater (>50% in 2022). We hope that readers will be interested to see the definitive dataset that quantifies microbiologically confirmed infections; to date only notification data have been available on line.

- (ii) that the upsurge of GAS due to M1-UK was limited to the UK (no evidence that similar events due to M1-UK have occurred in other geographic regions)

The upsurge due to M1_{UK} is not limited to the UK, but appears to have affected several European countries. This is now clarified (line 245). National data supported by genomic analyses from Belgium and Portugal demonstrate that nationwide late 2022/2023 upsurges were accounted for in large part by M1_{UK}. Although not including genomic analysis, the Netherlands (where most *emm1* strains are M1_{UK}) reported >50% of iGAS/STSS cases to be *emm1* in the late 2022 upsurge. More recent German ICU data suggests a similar upsurge in M1_{UK} (<https://pubmed.ncbi.nlm.nih.gov/38064158/>). Currently >70% of paediatric iGAS cases in Ontario are due to *emm1*, albeit genomic data are not available. https://www.publichealthontario.ca/-/media/Documents/I/2023/igas-enhanced-epi-2023-2024.pdf?rev=f16466608245457a984dcfa738930ad4&sc_lang=en

- (iii) lack of a mechanism for the observed increase in M1-UK

In terms of mechanism for the increase in paediatric infections, we believe that suppression of development of immunity (during the period of COVID-19-related NPI) will have had greatest impact in younger children, assuming that adaptive immunity develops in the early years through incremental exposures to *S. pyogenes*. Older children and healthy adults will presumably have retained some prior adaptive immunity. In England, the incidence of iGAS was greatest in the elderly (75 and over) and children aged 1-4 (9.8 per 100,000). The question regarding pleural infection is addressed below.

Specific comments:

1. Results, line 145: "pleural samples compared to [M1-global]" – are the sources/metadata harmonized between the two populations over time? Also, how significant is misclassification bias (i.e., patients with GAS from blood but with pulmonary/pleural disease and no pleural sample) in either dataset? These may impact the comparison made.

Isolate submission processes and metadata collected have been constant throughout the study period; all isolates from iGAS cases are submitted to our national reference laboratory with a standardised format. The isolates are classified by source (pleural, blood, csf joint fluid etc) and there was no change in this process. Furthermore, clinical practice regarding pleural aspiration and drainage has not altered during the study period; for children this is undertaken in tertiary referral centres and antibiotics have invariably been administered prior to pleural aspiration or drainage thereby reducing the isolates available. As such we fully accept that the number of isolates associated with pleural empyema will be lower than the number of clinical cases of empyema, however there is no reason to believe this would be different over time or between the two *emm1* lineages. We have clarified this in the Methods lines 364-6.

2. General: Association with pediatric and pleural samples – while statistically significant, there is generally a lack of mechanism for this association. It is possible, and in this reviewer's opinion highly likely, that the increased frequency of respiratory viruses during this same time period is a major contributor to the observed association (pediatric AND pleural isolation). Do the authors have data regarding concomitant respiratory viral infection/disease? It seems less likely, based on data presented on the post-pandemic M1-UK strains, that there is a bacterial genetic/phenotypic factor that contributes to this association but has not been tested.

We concur with the Reviewer that any association with empyema may be related to timing of the upsurge and have removed mention of empyema from the abstract. Lines 267-270: We acknowledge in the paper that concurrent respiratory viral infection may have contributed to empyema cases observed in the UK and clarify that it was the emergent clades that were associated with pleural samples.

Unfortunately we are unable to examine the specific association with respiratory viral infection in the current study because *S. pyogenes* isolates from invasive infection were submitted to the reference laboratory linked to isolate source and demographic data only. Respiratory viral PCR testing is increasingly undertaken as a point of care test and may not be reported to the reference laboratory unless done in a virology laboratory. Extracting information from other data sources could be undertaken in a separate study with linkage to NHS number, however virology data prior to 2022/23 are unreliable as respiratory viral PCR testing only became routine following emergence of COVID-19. We have inserted a comment to explain this in the Discussion (lines 269-273).

3. Ext data Figure 3 – No explanation of where these strains are derived from the larger population of M1-global or M1-UK. Of what are they representative?

We apologise for the lack of explanation regarding the ten strains used in the in vivo experiment. These were selected from 2022, chosen to be representative of each lineage (M1_{UK} and M1_{global}) and, in the case of M1_{UK}, to be representative of the three larger clades. Specifically we excluded any isolates with pre-existing *covS* or *covR* mutations. For the Reviewer's interest these isolates are shown on the phylogenetic tree below (now included as panel A in Extended data figure 3) and the selection process is now described in more detail in the Methods.

4. Discussion, lines 254-256: Agree that “throat specialists” and the mode of transmission is/was a critical factor during the pandemic GAS disease decline and post-pandemic surge. This is supported by the data presented by Aboulhosn et al. (PMID 37011014) that observed a similar disproportionate post-pandemic spike in *emm12* GAS.

Thank you; as Reviewer 2 suggests, it may be that any successful ‘throat specialist’ strain is likely to expand if there is an increased susceptible population. *emm12* account for a large proportion of circulating group A streptococci (i.e..pharyngitis or total GAS); we now cite the Aboulhosn Houston paper as an example of another location where *emm12* demonstrated an upsurge. Although *emm12* was associated with initial upsurges in the UK and Portugal (refs 15 & 29), these were multi-lineage upsurges. The arrival of new clones may impact the strains that cause epidemics; whether M1_{UK} has expanded beyond the nine US states reported in the recent CDC study (https://wwwnc.cdc.gov/eid/article/29/10/23-0675_article) is as yet unknown.

5. Discussion, lines 324-325: “raising the question of how such strains are maintained during periods of low [GAS] activity.” Undoubtedly, this is due to asymptomatic carriage and any number of data/studies are available to support this.

Thank you, we will amend the statement to “raising the question of whether strain-specific survival advantages exist during periods of low *S. pyogenes* population activity”

6. Discussion, lines 326-329: This is a known phenomenon called the “divorce effect” (PMID 33075052).

Thank you; we now cite this article in the section where we discuss immunity. Others have coined the terms ‘immunity debt’ or ‘immunity gap’.

Other changes-

We have updated the affiliations of authors from UKHSA due to organisational changes.

Two new references mentioned by Reviewer 2 have been added

We have updated the extended data file and added a panel to Ext Data figure 3.